# Sinus Lift Associated with Leucocyte-Platelet-Rich Fibrin (Second Generation) for Bone Gain: A Systematic Review

**DOI:** 10.3390/jcm11071888

**Published:** 2022-03-28

**Authors:** Ada Isis Pelaez Otero, Juliana Campos Hasse Fernandes, Tiago Borges, Leonardo Nassani, Rogerio de Moraes Castilho, Gustavo Vicentis de Oliveira Fernandes

**Affiliations:** 1Faculty of Dental Medicine, Universidade Católica Portuguesa, 3504-505 Viseu, Portugal; adaisis@yahoo.es (A.I.P.O.); tiagoferreiraborges@gmail.com (T.B.); 2Private Clinic, 3505-606 Viseu, Portugal; juchfernandes@yahoo.com; 3Center for Interdisciplinary Research in Health (CIIS), 3504-505 Viseu, Portugal; 4Division of Restorative and Prosthetic Dentistry, The Ohio State University College of Dentistry, Columbus, OH 43210, USA; nassani.1@osu.edu; 5Department of Periodontics and Oral Medicine, University of Michigan, Ann Arbor, MI 48109, USA; rcastilh@umich.edu

**Keywords:** sinus lift, sinus augmentation, platelet-rich fibrin, second generation

## Abstract

The purpose of this systematic review was to analyze sinus lifting procedures and to compare the efficiency of this treatment associated with the second generation of platelet-rich fibrin related to its effects on bone gain and to clarify the regenerative efficacy in sinus lift procedure, whether alone or as a coadjutant to other bone graft materials. The PICOT question was, “In clinical studies with patients needing a maxillary sinus lift (P), does the use of PRF either alone (I) or in conjunction with other biomaterials (C) improve the clinical outcome associated with bone gain and density (O), with at least three months of follow-up (T)?” An electronic search was conducted in the MEDLINE (PubMed), Science Direct, and Scopus databases through a search strategy. A total of 443 articles were obtained from the electronic database search. Sixteen articles met all criteria and were included in this review. Within the limitation of this study and interpreting the results carefully, it was suggested that a higher risk for implant failure after a sinus elevation might be seen in patients with residual bone ≤4 mm, and PRF application was effective, suggesting reducing the time needed for new bone formation.

## 1. Introduction

After tooth removal, the resorption of the alveolar ridge arises, which is particularly noticeable in the maxilla due to the thin cortical-type bone. Consequently, the maxillary sinus tends to be pneumatic, increasing in volume, hampering the implant positioning, and, in many cases, within a short period after tooth extraction [1,2]. This insufficient amount of bone makes it almost unmanageable to place an implant in the edentulous space, reducing rehabilitation possibilities [3]. Thus, the need for reconstruction and elevation of the maxillary sinus arose when dental implants became a standard procedure for dental rehabilitation.

Nonetheless, sinus elevation has not been an easy task, primarily due to the anatomical variations that may be present in the patient, such as a deviated septum or twisting of the uncinate process, since this might compromise the sinus integrity due to drainage lessening [3]. In addition, differences in the Schneiderian membrane, with a prevalence of 68%, and the variations of bone thickness, may allow perforations of the membrane affect the integrity of the surgery [2,3]. This is also true in crestal sinus lift procedures whereby, after corticotomy, the osteotome/chisel is pushed by a hand mallet to provoke a greenstick fracture of the bone with a space-making effect that could be filled with biomaterials or with dental implants at the same time. Nonetheless, it can be concluded that the crestal technique may create perforations in the MS membrane, which would be imperceptible without an endoscope device [4]. Furthermore, a device using magneto-dynamic technology can exploit the physical principles of electromagnetism to apply controlled forces on the local bone, minimizing the time of impact trying to avoid membrane perforations [5].

In order to reconstruct and prepare the region (maxillary sinus) to be rehabilitated, the search for a good regenerative option has never stopped [2]. Several grafting materials and procedure modifications have been developed to improve outcomes, reduce patient discomfort, and improve biocompatibility [6]. More than a few graft materials have been commonly used in maxillary sinus augmentation—most frequently, xenografts. They are widely available, biocompatible, have a low degradation profile, and have limited osteoconductive potential [2,6,7]. In addition, allograft and alloplastic are also utilized. However, they may have specific unfavorable characteristics, which are not present in the xenografts, making them the primary biomaterials in grafts used for a sinus lift [1,3]. Moreover, studies were carried out on guided bone regeneration (GBR), often using a collagenous barrier to segment tissues to enhance bone formation, such as that is normally applied in sinus augmentation procedures.

Intending to develop protocols that promote hemostasis and speed up the healing phase, fibrin has been the target of numerous studies. This biomaterial can be readily available and affordable for both patients and professionals [1]. The application of fibrin as an adjunct biomaterial has proven to be useful in numerous studies since it can enhance the concentration of growth factors in the surgical area, improving healing [1,3,7].

Based on those premises, the first generation of autologous platelet concentrates (APC) initially exhibited success in clinical applications [7]. Nonetheless, it had some barriers: costly, operator-dependent, an astringent within the tube, and extended generation time [1,6,7,8,9]. These features were surpassed by the second generation of APC, also known as platelet-rich fibrin (PRF), developed by Choukroun et al. in the early 2000s. This biomaterial has a greater amount of fibrin, platelet, and leucocytes. Either fibrin alone or with different materials have been utilized as a platform for essential cells to regenerate adipose, bone, sensorial, connective tissues, skin, ligaments, and tendons [1,8], thereby proving fibrin as a flexible natural biopolymer, showing an extraordinary potential in tissue recovery and wound-mending [8]. This generation did not use anticoagulants or any additional substances inside the tubes [1,9].

Moreover, the third generation of APC appeared in 2006, also known as the concentrated growth factor (CGF). It is rich in fibrin and growth factors, similar to PRF, which also help cell proliferation and differentiation, induce bone and tissue formation, and have pro-angiogenic properties [8]. Studies found similar results between the second and third generations [7,8,9,10,11].

This PRF clot contains a thick fibrin fiber arranged where platelets and leucocytes are snared, and it can serve as a system for other sorts of cells. Its affluence in leucocytes and platelets comes about in a steady discharge of growth factors such as platelet-derived growth factor (PDGF), transforming growth factor-beta (TGF-α), and vascular endothelial growth factor (VEGF), and insulin-like growth factor (IGF) [1,6,7,8,10,12,13]. In addition, studies have proven the efficacy of APC for wound healing and tissue regeneration, such as in extreme cases of chronic cutaneous ulcers [14]. Another study demonstrated the platelet concentrate effect to treat lesions, suggesting that PRF was proven to be suitable for lesion management while allowing for a reduction in systemic corticosteroid therapy, which is easily prepared, has a low cost, and can be used as an optimal scaffold for tissue healing processes [15]. Thus, there are potential benefits in combining platelets and fibrin matrices, employing a biomaterial for continuous release of growth factors, cell proliferation and differentiation, angiogenesis, antimicrobial effect, and remodeling [14]. The cost-effectiveness ratio of PRF is favorable by obtaining elastic and hemostatic fibrin from the patient’s blood without concerns about rejection [14,15].

APC has proved helpful in sinus lift surgery, helping release cytokines, thus accelerating neovascularization and controlling the inflammatory process. Correspondingly, the high concentration of growth factors, leukocytes, and platelets present on PRF matrices help develop new bone [3,8]. PRF alone as a biomaterial for sinus floor elevation has been reported. Aoki et al. (2016) [12] used PRF alone since it has therapeutic properties and serves to preserve the sinus membrane during subsequent placement of the implant and reduced the likelihood of sinus infection, which significantly increased a new bone formation histologically [12]. Other researchers have described similar outcomes by an average rate of improvement for the implant site with APC, but less so when combined with other biomaterials [2,3,4,6,9,10,11,12]. In addition, the combination of PRF and GBR technology affects repairing bone defects and can reduce patients’ pain during the repair process [16].

Since PRF is easily available, there is no need for a secondary surgical site, and it is not costly. It could be a good option as a sole biomaterial or adjunctive with a bone graft for the maxillary sinus lift procedure. Thus, the purpose of this systematic review was to analyze the available literature involving clinical studies that performed sinus lifting procedures and to compare the efficiency of second-APC generation related to its effects in bone gain, and to clarify the regenerative efficacy of PRF in sinus lift procedures, whether alone or as a coadjutant to other bone graft materials.

## 2. Materials and Methods

This systematic review was conducted following the Preferred Reporting Items for Systematic reviews and Meta-Analysis (PRISMA) guidelines [17], with the focused question being determined according to the Population, Intervention, Comparison, and Outcome, Follow-up (PICO) strategy.⁠ The protocol for this systematic review was registered on PROSPERO, CRD42021236993, provided by the Centre for Reviews and Dissemination/CRD (University of York).

### 2.1. Focused Question

The focused question for the present review was as follows: In clinical studies with patients needing a maxillary sinus lift (P), does the use of PRFs (second generation) either alone (I) or in conjunction with other biomaterials (C) improve the clinical outcome associated with bone gain and density (O), with at least three months of follow-up (T)?

### 2.2. Information Sources and Search Strategy

An electronic search was conducted by two reviewers (A.I.P. and G.V.D.O.F.), independently, electronic and manual search, through the MEDLINE/PubMed, Science Direct, and Scopus databases with a platform-specific search strategy combining terms (Table 1). An additional manual search was performed over the references of articles that met the inclusion criteria to identify relevant publications. Only articles published in the English language from January 2006 until and including August 2020 were included. If there was any doubt about whether or not a publication should be included, a third reviewer was consulted (T.B.).

### 2.3. Inclusion and Exclusion Criteria

The inclusion criteria for the selection were clinical studies, without the restriction of the period for follow-up and sample size, published in the English language from January 2006 to August 2020, which included histological and/or radiological evaluation, and only the second generation of APC or superior was used either alone or in conjunction with other biomaterials.

The exclusion criteria included articles that did not meet the inclusion criteria, in vitro studies, animal studies, serial studies (considered only the last publication), abstracts, posters, editorial letters, systematic/narrative reviews, and meta-analyses. Duplicate articles were also excluded, and the remaining pieces were screened first by title and then by abstract to evaluate if the inclusion/exclusion criteria were met.

### 2.4. Quality Assessment and Risk of Bias

We used a risk-of-bias evaluation tool, as described in the Cochrane Handbook for Systematic Reviews of Interventions. The risk of bias of each included that if the randomized controlled trial was assessed, was a low risk found when all parameters are received “+”? If received up to two “?”, this denoted a medium risk of bias; if at least one “-“ was received, the study was considered to have a high risk of bias. For non-randomized clinical trials, the Newcastle–Ottawa Scale (NOS) was applied to perform a methodological quality assessment. The observed total score between 1 and 3 was low quality; between 4 and 6 medium quality; from 7 to 9, high quality.

## 3. Results

Four hundred and forty-three articles were obtained from the electronic database search, with 67 on PubMed, 129 on Science Direct, and 247 on Scopus. An additional publication was considered from the manual search through the references of the included articles, including one more article. Of the 444 articles initially established and after careful screening by title and abstract, 33 studies were considered for full-text evaluation following the inclusion and exclusion criteria. After the assessment, 15 articles were considered by full text to be included in this systematic review (Figure 1). Due to not meeting the exclusion criteria detailed in Table 2, 18 full-text studies were excluded.

### 3.1. Studies Characteristics

Within the 15 studies selected for inclusion in this review, five were controlled clinical trials [18,19,20,21], six were randomized clinical trials [22,23,24,25,26,27,28], two retrospective clinical trials [13,29], one clinical-histologic study [28], and one case report [30]. These articles were published between January 2006 and August 2020.

Each study’s inclusion and exclusion criteria are detailed in Table 3. For all analyses, the inclusion was based on fit individuals. However, it was not completely specified in two of the 16 studies [19,20], and one of them only reported the enrollment of six healthy individuals [25]. All patients were over 18 years old and presented with edentulism of the maxilla with the necessity of sinus floor elevation and a small residual bone height.

The exclusion criteria were not specified in six of the 16 studies [13,20,25,29,30]. Therefore, the most common contraindications were the quantity of residual bone height, patients with disorders that might compromise the results, or contraindications for implant surgery. Perforation of the sinus membrane was a contraindication in one of the studies [21], and the amount of tobacco used was only specified in six of the studies. Smokers and ex-smokers were excluded in one study [23], a current smoking habit was part of the exclusion criteria in four studies [22,26,27,28], and heavy smoking was part of the exclusion criteria in one study [18] (Table 4). Diabetes was only specified as exclusion criteria in four studies [28,30,31,32], and alcohol and drug abuse were only specified in two studies [18,30].

The amount of residual bone height was referred to as inclusion and exclusion criteria in eight of the studies. Cho et al. denoted reduced residual bone height, which made the placement of implants with a length longer than 8.5 mm impossible as inclusion criteria, and less than 5 mm of residual bone height as an exclusion criterion. Maxillary atrophy with residual ridge < 5 mm was part of the inclusion criteria in three of the studies [26,27,28] and less than 4 mm in the other two studies [18,30]. Seven millimetres or less of residual bone height was a criterion for inclusion of the study of Kılıç et al. [24].

Chronic sinus infection and chemotherapy were also considered as exclusion criteria in six of the studies analyzed for this review [24,26,27,30,31]; the use of specific drugs such as bisphosphonates was only mentioned as an exclusion criterion in three studies [26,28,30]. Nizam et al. was the only study that presented the amount of time of being edentulous specified as exclusion criteria; in this case, an edentulous patient of less than one year was excluded from the research.

### 3.2. Quality Assessment and Risk of Bias

For non-randomized studies, only one out of nine studies had medium quality; all others (n = 8) achieved a high quality in the assessment (Table 4). Therefore, for randomized studies, no one reached a low risk of bias, with three studies obtaining medium risk (Gassling et al., 2013; Kılıç et al., 2017; Pichotano et al., 2019), and another three with a high risk of bias (Table 5).

### 3.3. Total Patients, Age, and Amount of Surgeries

These trials included 354 patients, 431 sinus lift procedures, and 683 implant placements, including patients ranging in age from 18 years old for Cho et al.’s younger patient to 90 years old for Toffler et al.’s senior patient [20,22]. Because some studies only reported the youngest or/and oldest patient who had been part of the trial, and others did not report the total number of patients, the mean age of the treatment group could not be computed. These findings will be discussed in greater depth subsequently.

Cho et al. reported their patient niche as any healthy patient 18 years of age or older. They had 40 patients, 21 males and 19 females, and placed 45 implants in 40 lifted sinuses [22]. Kempraj et al. performed 22 direct sinus lifts in 11 patients, with an age range between 20 and 60 years old; they did not report how many female or male patients were evaluated and did not report the placement of implants. Pichotano et al. (2019) had the mean patient age of 54.17 ± 6.95 years, from 43 to 63 years, comprised of a total of 12 patients that needed bilateral sinus lift (six male and six female), and placed a total of 38 implants; in their 2018 report, they had a male patient of 59 years old in need of bilateral sinus elevation and to whom two implants were placed.

Aoki et al. (2018) had the biggest age range with patients as young as 29 and as old as 82, with a mean age of 57.6 years. They had a sample of 17 females and 17 males (34 patients) who received 34 sinus lift procedures and 71 implants placed [29]. Their report (2016) had two different patients, a 28-year-old female who needed sinus elevation and implant placement and a 58-year-old male with a failed implant who needed sinus elevation for posterior implant placement [13].

Nizam et al. reported a patient mean age as 49.92 ± 10.37, with a range from 35 to 65 years old, comprised of nine male patients and four females; they realized 26 sinus lift surgeries placed a total of 58 implants in those 13 patients described [26]. Kılıç et al. had a total of 26 patients divided between the different test and control groups, with a mean age of 49.92 ± 10.37, two females and seven males in the control group; four females and five males in the P-PRP group and three females and five males in PRF group; they performed a total of 26 sinus lift, and no implant placement was reported [24].

Gassling et al. reported 12 sinus lift surgeries and 32 implants placed in six patients with a mean age of 61 years old; the number of females or males was not reported [25]. Tatullo et al. reported the placement of 240 implants on 60 patients, 48 females and 12 males, with an age range of 43 to 62 years old, after 72 sinus lift surgeries [28].

Zhang et al. and Olgun et al. specified the age range of the control and the test group. Zhang et al. test group mean was 46.2, and the control group was 43.5, while Olgun et al. reported it as 53 ± 8.96 and 51 ± 7.94, respectively. Zhang et al. performed 11 sinus elevations in 10 patients (two females and eight males) and placed 11 implants. On the other hand, Olgun et al. placed 37 implants after sinus lift surgery on 18 patients (nine females and nine males) [27,33].

Toffler et al. reported the most patients (n = 110) and the majority of sinus lift procedures (n = 138). They placed 138 implants in all, with 70 women (92 sites) and 40 men (46 sites). The patients’ ages ranged from 34 to 90 (mean age of 58.4 years) [20]. Conversely, Choukroun et al. and Anitua et al. did not report the age range or the proportion of female or male patients treated. Indeed, Choukroun et al. did not even report the number of patients, reporting nine sinus lift surgeries followed by nine implants placement, and five patients and five sinuses lifting reported [21,31].

### 3.4. Follow-Up

The follow-up period and frequency varied between authors. The shorter follow-up was three months and the longer, seven years. Aoki et al. had an average follow-up time of 3.43 years in one of his studies, from one to seven years [29] and 24 months on other of their studies. Nizam et al. and Kılıç et al. had a total follow-up of 18 months [24,26]. Olgun et al., Toffler et al., Gassling et al., and Cho et al. reported a follow-up period of one year [20,22,25,27].

A follow-up period of 8 months was described by Tatullo et al. and Choukroun et al. [28,31] and of 6 months by Zhang et al. [33]. Kempraj et al. recounted several follow-up periods from 3 months to over 2 years; Pichotano et al. of 10 months and Anitua et al. of 33 months [18,21,30], as can be seen in Table 6.

### 3.5. Surgical Approach and Residual Bone Height

The most common surgical approach among all authors was the lateral window approach [21,25,26,28,30,31,32,33], modified Caldwell-Luc was only reported by Kılıç et al., and the balloon-lift technique is only mentioned by Olgun et al. [24,27]. The authors’ other most common surgical approach for sinus lifting was the crestal, trans-crestal, and mid-crestal approach [13,18,20,22,23,29], as detailed in Table 6.

### 3.6. Membrane Perforation and Implant Failure

Four authors reported complications such as membrane perforation, and only two reported implant failures. Kempraj et al. reported a membrane perforation that was healed by adding a PRF membrane [18]; Kılıç et al. reported five, Toffler et al. described three, whereas Choukroun et al. stated one sinus perforation [20,24,31].

Toffler et al. reported three implant failures before loading [20], and Aoki et al. reported seven implant failures—all of which occurred when the residual bone height was less than 4 mm [29].

### 3.7. Biomaterials Association

All of the studies used PRF as the main biomaterial; some only used PRF, while others compared the effectiveness of using only PRF or PRF combined with other biomaterials, such as Bio-Oss^®^, β-TCP, or freeze-dried bone allograft.

The most common adjunct biomaterial used by the researchers was Bio-Oss^®^ (Geistlich Pharma AG, Wolhusen, Switzerland). The study by Kempraj et al. was the only one that compared sinus augmentation with only PRF and PRF + Bio-Oss^®^ [18]; Bio-Oss^®^ with PRF compared to Bio-Oss^®^ alone was preferred by five authors [21,23,26,28,33].

A mixture of Bio-Oss^®^ with cortilocancelous bone and PRF for the experimental group and cortilocancelous bone with Bio-Oss^®^ and collagen membrane for the control group were the biomaterials preferred by the study of Gassling et al. [25]. Pichotano et al. (2018) compared the use of a collagen membrane with a PRF membrane and Bio-Oss^®^ versus only using a collagen membrane and Bio-Oss^®^ in the control group [30].

Choukroun et al. and Kılıç et al. did not utilize Bio-Oss^®^ as a bone substitute material. Still, a freeze-dried bone allograft was the preferred biomaterial by the first, and PRF with β-TCP compared to PRP with β-TCP and β-TCP alone was preferred by the latter [24,31].

Four authors [13,20,22,29] preferred not to use any type of bone substitute but just PRF membranes as their preferred biomaterial; of those authors, only Cho et al. compared the use of PRF versus saline solution in the sinus augmentation procedure (Table 7).

#### 3.7.1. PRF as the Sole Biomaterial

PRF was employed as the only grafting material in five studies [13,20,22,27,29], totaling 232 sinus lift surgeries in 204 patients (crestal approach), and 293 implants were inserted in the enhanced areas. PRF membranes were inserted into the sinus cavity to fill the space. The average residual bone height at T0 was 5.45 mm, spanning from 0.56 to 9.6 mm. The average of new bone formed with PRF alone cannot be stated since Aoki et al. (2016 and 2018) did not address it.

For this group of analyses, Aoki et al. (2018) reported seven implant failures, but exclusively in patients with a residual bone height less than 4 mm at T0 [29]. Toffler et al. had three implant losses, all of which occurred before the loading phase in areas with a sinus floor elevation with a gain of 3 to 4 mm but an initial residual bone height of 4 mm. All implants were replaced 16 weeks later with no further complications [20].

During the recovery process, no significant complications were identified. Evident sinus membrane perforations were identified in five cases (3.5%) [20] and were easily repaired using PRF membranes.

#### 3.7.2. PRF with Allograft

Only Choukroun et al.’s research [31] utilized PRF combined with freeze-dried bone allograft (FDBA). In nine cases, sinuses were executed using the lateral approach. They filled three sinuses that only acted as a control group with FDBA granules (Phoenix, TBF, Mions, France). The allograft/PRF mixture was used to fill the remaining six sinuses (test group), and PRF membranes were employed to cover the sinus membranes. For implant placing, second-stage surgery was executed, and samples were taken four months after, for the experimental group and eight months following surgery for the control group. Perforation of the Schneiderian membrane occurred in one case, but it was easily repaired using PRF [31].

In the allograft/PRF group, a histomorphometric examination revealed 65% of vital new bone and 35% of inert bone within the bone trabecular areas after four months. The proportion of vital new bone and inert bone visualizing the trabecular bone areas was 69% of vital new bone and 31% inert bone in the control group after eight months [31].

As per the histomorphometric assessment, the test group (FDBA+PRF) and the control group (FDBA alone) seemed to have similar bone structures. Nonetheless, the healing times of the two groups were not the same—four versus eight months, respectively; it appeared that when PRF was combined with FDBA to perform sinus floor augmentation, bone regeneration appeared to be enhanced, enabling implant insertion after only four months of recovery [31].

#### 3.7.3. PRF with Xenografts

PRF was used in combination with Bio-Oss^®^ (Geistlich Pharma AG, Wolhusen, Switzerland) in nine studies with 124 patients and 191 sinus lift surgeries; the lateral approach was the preferred surgical access method, performed by seven of the researchers [21,23,25,26,28,30,33]. Only Kempraj et al. performed a mid-crestal surgical procedure, with 381 implants placed in 157 elevated sinuses, meanwhile, without any implant placed [18].

A radiographic examination was done using panoramic radiography or a CBCT scan to assess bone development in all trials. No significant complications were identified during the recovery process. Only Kempraj et al. reported a perforation of the Schneiderian membrane, which they solved by placing a PRF membrane and then waiting three months to elevate the sinus again [18].

The exact amount of bone gain was not reported by all researchers. Some relied only on the X-ray bidimensional images to demonstrate a significant amount of bone gain [26,28,30]. The authors that reported a higher amount of bone gain were Anitua et al. (test group varying from 20% to 30% versus just 8% gain in the control group) [21], Kempraj et al. also reported a significant difference between the control (PRF alone 6.545 mm) group and the test group of PRF + Bio-Oss^®^ reporting an average gain of 12.636 mm [18].

#### 3.7.4. PRF with Synthetic Bone Graft

PRF was used in conjunction with beta-tricalcium phosphate (β-TCP) in one of the included studies [24], including a total of 26 patients who were divided into three groups, the control group (nine patients), PRP with a β-TCP group (nine patients), and PRF group (eight patients). PRF was prepared using the procedure described by Choukroun and was mixed with the β-TCP particles by hand. Then, this mixture was placed in the correspondent groups for sinus augmentation. After surgery, the lateral window was covered with a collagen membrane in all groups [24].

All patients were recalled for follow-up and implantation six months after the sinus elevation, and bone biopsies were harvested. The composition and distribution of histologic features in the P-PRP, PRF, and control groups biopsies were remarkably similar [24]. They all displayed a large amount of new bone development and angiogenesis around the β-TCP units. Many inflammatory cells were found in the PRF group compared with the control or P-PRP groups. The quantity of new bone formation in the PRF group was 32.03%, resulting in a slightly lesser development than when mixed with PRP [24].

### 3.8. New Bone Formation

The amount of new bone formed was not reported in six studies [13,26,28,29,30]. All the other studies reported the quantity of bone gain in different ways; some reported as a percentage and others by the number of millimeters. The highest amount of bone gain of 33% was reported by Kılıç et al. However, the authors did not report a difference between the control and the test group [24]. Anitua et al. reported the second-highest amount of bone gain (20–30%) in the test group, much better than the amount of new bone formation reported for the control group (only 8%) [21]. Kempraj et al. reported the highest quantity of bone gain when PRF was used in aggregation with Bio-Oss^®^, with an average bone height gain of 12.636 mm, compared to the mean of 6.545 mm obtained when PRF was applied by itself [18]. Toffler et al. reported a mean increase of 3.5 mm of bone when using only PRF [20]. All other studies reported a similar bone gain, comparing the control and test groups (Table 8).

### 3.9. Implant Stability Quotient (ISQ)

ISQ was only reported by four of the studies included in this review [23,27,28,30]. This measurement may be taken in two ways, using implant resistance (Periotest, Avtec Dental, Mount Pleasant, SC, USA) or resonance frequency analysis (RFA) [34]. This measure is important to evaluate the dental implant’s stability and could be measured from 1 to 100. However, a range of 55 to 85 is considered suitable to declare implant stability and comprehensive development of osseointegration [30,34].

Pichotano et al. reported the ISQ value in their research [23,30]. The method to analyze the implant stability rate was the Osstell^®^ device (Integration Diagnostics, Gothenburg, Sweden). It was done immediately after implant placement and at the loading phase for both groups [23,30]. In their 2018 study, the authors measured ISQ at four months for the test group and eight months for the control group, finding an ISQ of 75.13 ± 5.69 in the control group, which was significantly higher than that found on the test group 60.90 ± 9.35. Nonetheless, at the time of loading, the ISQ value of the test group had incremented significantly to 76.08 ± 5.86, reducing the previous difference. They reported no significant difference between both groups for this analysis [30].

Pichotano et al. (2019) followed a similar procedure to measure the ISQ. The authors used the same type of RFA device, Osstell^®^ [23]. An initial ISQ mean in the test group was reported as 69.5 at four months after sinus elevation. An ISQ value of 77 was on the control group at the initial measurement eight months after the initial surgery. A second measurement was taken at the loading phase, which resulted in an ISQ average of 81.5 in the test group, which demonstrated a significant boost when compared with the initial measurements, and of 75.75 in the control group, displaying a similar result than on the initial ISQ [23].

Olgun et al. only measure the ISQ three months after implant placement using the Ostell^®^ device [27]. They reported an ISQ value of 68.50 in the test group and 66.37 in the control group, showing no significant difference between groups [27].

Tatullo et al. also used the resonance frequency analysis (Osstell^®^) to assess the implant stability rate. The results were lower than commonly found, 37.2 in the early protocol group (106 days after sinus lift), 36.8 in the intermediate protocol group (120 days after sinus lift), and 39.1 in the late protocol group (150 days after sinus lift) showing no significant difference between the test and control groups [28].

## 4. Discussion

This study aimed to verify whether the second generation of APC (PRF) affected bone formation in maxillary sinus augmentation surgeries. This review revealed that just a few clinical investigations had been conducted. Furthermore, given the diversity of the identified research in terms of surgical technique, grafting material, implant placement time, protocol, sample characteristics, biopsy, implant placement healing time, and follow-up period, the development of meta-analysis was not possible.

Bone graft biomaterials were employed as space maintainers and bone scaffolds during sinus lift to enhance healing in the sub-sinus area. The widespread consensus was that several biomaterials could be used in sinus lift procedures due to the high osteogenic potential of the Schneiderian membranes [35]. The use of PRF (second generation) during sinus lift, even without bone substitute, appears to be a highly attractive choice, particularly for sinus membrane protection [1,6,9,11,13,21,23,28,35]. Moreover, it has been considered a low-cost and straightforward technique employed in daily practice [1,2,21,28,33], creating an autologous fibrin membrane [1,9].

This study did not accept the first generation because it did not have a membrane structure, a greater difficulty in terms of handling, thus creating bias concerning the efficiency and outcomes. Moreover, it was proved through in vitro studies that PRF releases autologous growth factors gradually and expresses a stronger and more durable effect on the proliferation and differentiation of osteoblasts than PRP [36]. In addition, the third generation was not included (CGF) [9], which is composed of a denser fibrin matrix and is thus richer in growth factors. This is due to a lack of significant differences between second and third generations, with no explicit agreement in the literature, which has contradicted the literature in clinical results [37]. This may be due to the reality that, indeed. However, it has been demonstrated that both concentrates have a great discharge potential of growth factors and cytokines; there is not a clear understanding within the literature as to CGF representing so much progress as to be called the third generation. It is being utilized interchangeably by clinicians with comparable results [7,8,9,10,11,38].

Based on the literature, the use of PRF surpassed the benefits of the use of the first generation of ACPs, as reported by Batas et al., whose study demonstrated that adding platelet-rich in growth factors (PRGF), first-generation, associated with Bio-Oss^®^ grafting for maxillary sinus floor augmentation, did not improve the bone growth six months after surgery when compared to bone grafting alone. The authors reported that PRGF as an adjuvant to Bio-Oss^®^ for maxillary sinus lift did not appear to enhance or interfere with bone development inside the human sinus six months after surgery, except for improved handling, during the procedure [32].

### 4.1. PRF and Other Biomaterials

Thus, biomaterials such as L-PRF can be used either as a single graft material or combined with other materials [9,21,23,26,28,30,31]. In clinical practice, the Bio-Oss^®^ associated with L-PRF is frequently used. Bio-Oss^®^ is a biocompatible and osteoconductive biomaterial, and various studies have shown that it has a high clinical success rate with satisfactory results [19,21,23,26,28]. However, once Bio-Oss^®^ has a lack of osteogenic and osteoinductive characteristics, as well as the fact that maturation can take up to eight months, the implant placement limitation is established [14,20,28]. As a result, it is critical to include additional biomaterial that can act as a scaffold, allowing osteogenic cells to migrate to the graft, permitting bone neoformation [1,13,19,26,28]. Therefore, mixing of L-PRF and Bio-Oss^®^ occurs in the existence of a fibrin network, which minimizes the dispersion of graft particles, resulting in less difficulty to insert biomaterials [19,23,26].

Another study by Choukroun et al. used PRF with a freeze-dried bone allograft (FDBA) to elevate the sinus floor. Histological assessments suggested that sufficient new bone growth was seen in their study. Furthermore, the blended graft material has the potential to speed up the healing process. According to histomorphometric analysis, the control group (FDBA alone) and the test group (FDBA + PRF) had similar bone characteristics. Even though the two groups’ healing durations were not equal, four for control and eight months for the test group, it indicated that when PRF is paired with FDBA to perform sinus floor augmentation, bone regeneration appears to be improved, allowing for implant insertion after only four months of healing [31]. Similar results were obtained by Pichotano et al. and Tatullo et al., who found a faster maturation of the bone graft, which predicted a shorter healing time (four months) before implant placement and 100% of implant survival rate [28,30].

### 4.2. Only PRF Inserted

PRF membranes may provide excellent results—usually when the sinus is elevated and has a residual bone height of more than 4 mm. Most reported implant failures were in maxillary sinuses whose initial residual height was below those parameters [29]. Thus, PRF alone for maxillary sinus augmentation is not adequate whether the bone availability needed is over 3 to 4 mm [22] as the sinus membrane may collapse the PRF [18]. Hence, it is necessary to add other biomaterials and bone grafts to the PRF in those cases [18,22].

Another factor to consider is if a PRF membrane is going to be used without any other biomaterial, it could be recommended that the immediate implant placement act as a tent pole [20,29], which could help maintain the elevated membrane in the desired position. Then again, the most important factor for implant survival rate when using PRF, either alone or in conjunction with another biomaterial, is the initial residual bone height, since this may change the final prognosis [6,13,20,29].

Regarding bone gain or new bone formation, the literature agrees there is no significant difference in the amount of bone formation when only PRF is used than when PRF is used with other biomaterials [18,19,23,24,25,27,31]. Conversely, Anitua et al. reported an increase between 20% and 30% in the test group versus 8% in the control group [21].

### 4.3. Implant Stability Quotient (ISQ)

The main reason for a maxillary sinus lift is to be able to rehabilitate the area with implants, either immediately or at a second stage surgery, with a significant factor for decision directed to the professional’s choice, according to the quantity of residual crestal bone [6,11,20,29] in the posterior maxilla. Thereby, it is related to the possibility of establishing a high value of primary implant stability, which shows the importance of the ISQ. However, these data were only reported in four studies [23,27,28,30]. Similar ISQ quotients were reported by all studies ranging from 60 to 75 [23,27,30]. Negatively, Tatullo et al. reported an average for ISQ of 35 for all different groups [28]. Tatullo et al.’s results contradict and are not consistent with the literature [23,34].

### 4.4. Study Limitations

A direct comparison of published reports was difficult due to the wide range of study design, inclusion and exclusion criteria, as well as patient age and gender, smoking habits, implants placed or not, follow-up intervals, use or non-use of a PRF or collagen membrane to cover the graft, and amount of residual bone height between the sinus floor and alveolar crest. These demonstrated the difficulty experienced when attempting to draw findings from non-controlled trials due to the inclusion of many other confounding variables that may impact the success of the sinus lift treatment.

As a result, it was impossible to isolate all these distinct variables in this systematic review, but it was non-feasible to perform a meta-analysis. The quantity of bone developed after surgery and ISQ were explored as the primary outcomes since they were easy to determine. However, long-term success is more valuable yet involves other criteria. Regrettably, these criteria varied considerably between investigations.

## 5. Conclusions

Research showed that sinus augmentation treatment is a predictable treatment, which can achieve a high success rate, suggesting that a higher risk for implant failure after a sinus elevation may be seen in patients with a residual bone height of 4 mm or less. However, within the limits of this systematic study, such as the differences found in the studies included (wide range of study design, inclusion and exclusion criteria, patient age and gender, smoking habits, implants placed or not, follow-up intervals, use or non-use of a PRF or collagen membrane to cover the graft, and amount of residual bone height between the sinus floor and alveolar crest), the results must be carefully interpreted. Thus, for maxillary sinus without any lesion, the clinician must evaluate the amount of residual alveolar height and type of biomaterial who will utilize (PRF alone or in combination with other bone grafts), which may cause an impact on the outcome.

Moreover, the application of PRF either alone or in conjunction with another biomaterial has been suggested as an effective biomaterial, reducing the time for new bone formation and consequently, the time necessary for implant rehabilitation. Thus, it is suggested that the healing period can be shortened when PRF is used in the maxillary sinus lift elevation surgery. In addition, more high-level and standardized studies are needed to confirm PRF’s performance, alone or in combination, in the maxillary sinus elevation.

## Figures and Tables

**Figure 1 jcm-11-01888-f001:**
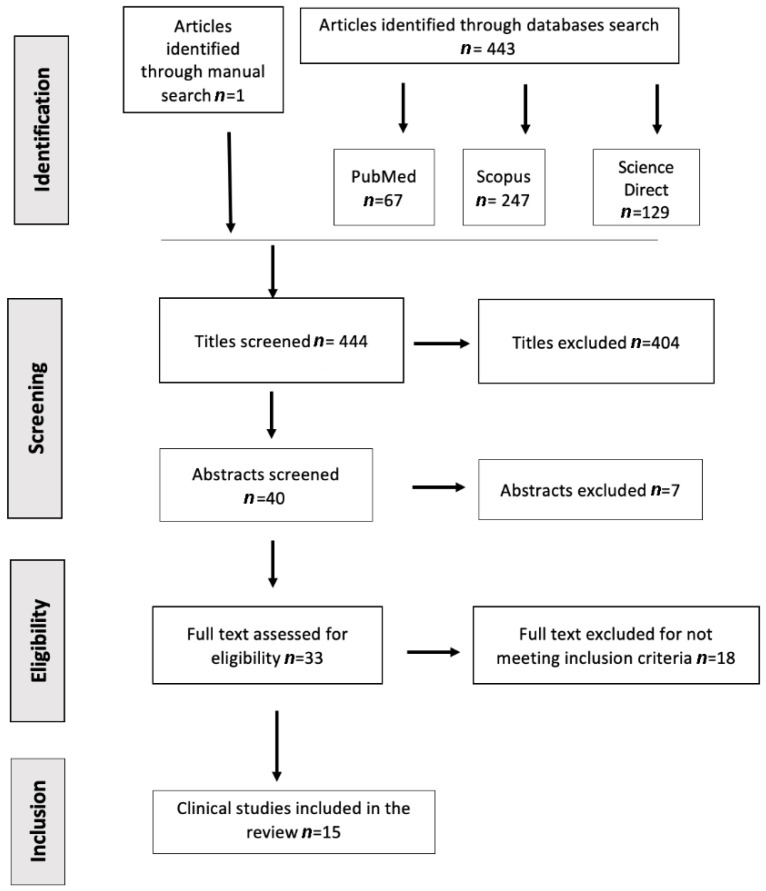
Flow diagram for the search strategy and selection process.

**Table 1 jcm-11-01888-t001:** The search strategy was carried out, and filters were applied.

**#1**	P—In clinical studies with patients needing a maxillary sinus lift
	((“Sinus Floor Augmentation” [mesh Terms]) OR (“Sinus floor elevation) OR (Sinus lift))
**#2**	I—Does the use of autologous concentrated platelets
	((“Platelet-Rich Fibrin” [mesh Terms]) OR (autologous concentrated platelets”) OR (Second generation; platelet concentrate) OR (L-PRF]) OR (PRF))
**#3**	C—with or without the addition of other biomaterials
**#4**	O—improve the clinical outcome associated with bone gain and density
**Search combination**	(#1 and #2) No combination was done with #3 and #4, since the preponderance of the papers on sinus elevation considers bone gain and density. The combination with keywords related to the outcome would further limit the results.
**Filters**	English, Humans, January 2006–August 2020, in vivo, Clinical Studies

**Table 2 jcm-11-01888-t002:** Excluded studies and reason for exclusions.

Author/Year	Reason for Exclusion
Xin et al., 2020	Data did not clear for evaluation
Silberman et al., 2020	Data does not meet inclusion criteria (focus on perforation)
Mohamadamin Damsaz et al., 2020	Review, Data did not clear for evaluation
Xie et al., 2019	Full text only available in Chinese
Wang et al., 2019	Data does not meet inclusion criteria (focus on infection)
Chandra et al., 2019	Data does not meet inclusion criteria (maxillary sinus)
Batas et al., 2019	Focus on first-generation
Öncü et al., 2017	Data does not meet inclusion criteria (focus on perforation)
Karaca et al., 2017	Only the first generation of ACP discussed
Peker et al., 2016	Data does not meet inclusion criteria (animal)
Anitua et al., 2016	Report of immediate placement
Taschieri et al., 2015	Only the first generation of ACP discussed
Tanaka et al., 2015	Data does not meet inclusion criteria
Anitua et al., 2015	Data mostly on short implants and first-generation
Troedhan et al., 2015	Only anterior maxilla discussed
Amin Rahpeyma, 2014	Only poster/abstract available
Inchingolo et al., 2010	Full text not available
Meyer et al., 2009	Full text only available in French

**Table 3 jcm-11-01888-t003:** Studies’ inclusion and exclusion criteria.

Author/Year	Type of Study	Inclusion Criteria for Each Included Article	Exclusion Criteria for Each Included Article
Cho et al., 2020	Randomized control trial	Any healthy patient over the age of 18 has edentulism in the posterior maxilla and a decreased RABH, making the placement of implants longer than 8.5 mm unfeasible.	Systemic or local contraindications for implant placement include a history of untreated metabolic problems, smoking habits, bruxism, or uncontrolled periodontal disease, as determined on a cone-beam computed tomography (CBCT) scan and a residual bone height less than 5 mm.
Kempraj et al., 2020	Clinical Study	Patients between 20 and 60 years old with severe maxillary atrophy in the sinus region less than 4 mm	Patients with uncontrolled systemic disorders, heavy smoking, alcohol or drug addiction, and uncontrolled periodontal disorders are at risk.
Pichotano et al., 2019	A double-blinded, randomized controlled trial	Patients having a residual bone height of less than 4 mm who needed bilateral sinus floor augmentation for implant placement in the posterior maxillary area (based on CBCT)	Patients with poor general health, smokers or ex-smokers, alcoholics and drug addicts, irradiated patients, pregnancy, and bisphosphonate therapy Immunosuppressive drugs, blood platelet abnormalities, and chronic pain Patients with sinusitis or other pathology in the maxillary sinus With diabetes that is uncontrolled
Aoki et al., 2018	Clinical Retrospective	Maxillary posterior tooth loss, good general health or manage medical conditions, implant placement by sinus floor elevation with PRF alone as the grafting material, informed consent granted, and follow-up visit performed at our facility following implant installation.	Not specified
Pichotano et al., 2018	Case clinical report	One patient split-mouth, the patient reported no relevant medical history that could compromise bone healing, denied smoking, or used alcohol	Not specified
Nizam et al., 2018	Prospective randomized clinical trial	Systemically healthy, age 21 years or older, implant therapy required in the bilateral posterior maxilla with a residual bone height of less than 5 mm, and periodontally healthy	Any systemic disease, use of any medications that could interfere with bone metabolism (i.e., corticosteroids, bisphosphonates), smoking, history of maxillary sinusitis or sinus surgery, history of reconstructive or previous implant surgery, and being edentulous for more than a year are all factors to consider.
Cömert Kılıç et al., 2017	Randomized clinical trial	Age > 20 years, atrophic maxilla, previous posterior tooth loss, residual bone crest height = 7 mm or less on orthopantomography, and atrophic maxilla.	Had a maxillary sinus infection or hematologic, neurologic, or systemic problems, had radiotherapy or chemotherapy, had inflammatory or connective tissue illness, or had a malignant disease in the head and neck region.
Aoki et al., 2016	A Clinical retrospective study	Case 1: good general health and non-smoker Case 2: no systemic pathology, smoker	Not specified
Gassling et al., 2013	Randomized controlled study	Six healthy patients	Not specified
Tatullo et al., 2012	Randomized clinical trial	A preoperative radiological and tomographic assessment revealed maxillary atrophy with a less than 5 mm remnant ridge. Due to toothlessness, anatomic-functional rehabilitation of the posterior maxilla is required.	Hemo-coagulative diseases Diabetes Incompetence/Immunological deficiency Previous head-neck radiation treatment Normal bone physiology anomalies Bisphosphonate-based treatments Smokers and ex-smokers would both be excluded
Zhang et al., 2012	Clinical Study	Not specified	Blood platelet problems, aspirin therapy before surgery, viral and metabolic diseases, radiation, and acute and chronic maxillary sinus inflammation are all things to consider.
Toffler et al., 2010	Clinical Study	Not specified	Not specified
Choukroun et al., 2006	Histologic, clinical study	Thrombocyte concentrations in the blood are within normal limits, and there is no history of maxillary sinus irritation. Significant atrophy of the maxilla was discovered during the clinical examination and preoperative radiography.	Patients with immunologic disorders, uncontrolled diabetes, current chemo- or radiotherapy, or a history of drug misuse should not be considered.
Olgun et al., 2018	Randomized clinical trial	Age ≥ 18 years Systemically healthy Non-smokers. Full-mouth plaque and bleeding score ≤15%; Presence of a residual crest height of <5 mm in the posterior maxilla as detected on X-rays.	Infectious and metabolic illnesses; Blood platelet abnormalities Chemotherapy or radiotherapy is still being administered. Chronic sinusitis in the maxillary sinuses is a common occurrence. Antibiotics and/or anti-inflammatory medicines are being taken.
Anitua et al., 2012	Clinical Study	All patients in the research had severe alveolar atrophy and a class D residual bone height.	Any local or systemic disorders can make the treatment ineffective. Perforation of the Schneiderian membrane.

**Table 4 jcm-11-01888-t004:** Quality assessment risk of bias for non-randomized clinical trials.

	Selection				Comparability	Outcome			Total Score
	Adequate Definition of Patient Cases	Representatives of Patient Cases	Selection of Controls	Definition of Controls	Control for Important or Additional Factors (Max of 2X)	Ascertainment of Exposure	Was Follow-Up Long Enough for Outcomes to Occur	Adequacy of Follow Up	
Kempraj et al., 2020	X	X	X	X	XX	X	X	X	9
Aoki et al., 2018	X	X	X	X	X	X	X	X	8
Pichotano et al., 2018	X			X	XX	X	X	X	7
Nizam et al., 2018	X	X	X	X	X	X	X	X	8
Aoki et al., 2016	X	X			X	X	X	X	6
Zhang et al., 2012	X	X	X	X	X	X	X		7
Anitua et al., 2012	X	X	X		X	X	X	X	7
Toffler et al., 2010	X	X	X	X	XX	X			7
Choukroun et al., 2006	X	X	X	X	XX	X	X	X	9

**Table 5 jcm-11-01888-t005:** Quality assessment risk of bias for randomized clinical trials.

	Random Sequence Generation (Selection Bias)	Allocation Concealment (Selection Bias)	Blinding of Participants and Personnel (Performance Bias)	Blinding of Outcome Assessment (Detection Bias)	Incomplete Outcome Data (Attrition Bias)	Selective Reporting (Reporting Bias)	Other Bias
Cho et al. 2020	+	?	?	?	+	+	+
Pichotano et al. 2019	+	+	?	+	+	+	?
Olgun et al. 2018	+	+	?	−	+	+	+
Cömert Kılıç et al. 2017	+	?	?	+	+	+	+
Gassling et al. 2013	+	+	?	+	+	+	?
Tatullo et al.. 2012	+	?	−	−	+	+	+

Table 5 shows the quality of the randomized studies that were included. Risk of bias summary for randomized studies (“+” denotes a low risk of bias; “?” denotes an unknown risk of bias; and “−” denotes a high risk of bias).

**Table 6 jcm-11-01888-t006:** Follow-up period and surgical approach.

Author/Year	Follow-Up	Surgical Approach	Residual Bone Height
Kempraj et al., 2020	3 months over 2 years	Mid-crestal incision	Less than 4 mm
Cho et al., 2020	1 year	Trans-crestal sinus lift	Over 5 mm
Pichotano et al.,2019	4, 8 months, and at implant placement and loading (time not specified)	Lateral window	Less than 4 mm
Pichotano et al., 2018	10 months	Lateral window	Not reported
Olgun et al., 2018	1 year	Balloon-lift technique	Less than 5 mm
Nizam et al., 2018	12 months after implant loading (18 months)	Lateral window	2.53 mm
Aoki et al., 2018	Average 3.43 years (1–7 years)	54 implants were placed by trans-crestal approach and 15 by the lateral approach	Ranged from 0.56 mm to 9.60 mm
Cömert Kılıç et al., 2017	18 months 6month after surgery 12 months after loading	Modified Caldwell-Luc	Not reported
Aoki et al., 2016	24 months	Trans-crestal	Pt#1 2.7 mm pt#2 less than 2 mm
Zhang et al.,2012	6 months	Lateral window	Less than 5 mm
Tatullo et al., 2012	8 months	Tatum’s technique (lateral window)	Less than 5 mm
Gassling et al., 2013	1 year follow up after implant placement (17 months)	Lateral window	Less than 5 mm
Anitua et al., 2012	33 months	Lateral window	Less than 3 mm
Toffler et al., 2010	1 year	Trans-crestal	4 to 8 mm
Choukroun et al., 2006	8 months	Lateral window	Not reported

**Table 7 jcm-11-01888-t007:** Biomaterials were applied within the studies.

Author/Year	PRF Alone	PRF + Bone Substitute	Only Bone Substitute
Kempraj et al., 2020	YES	PRF + BIO-OSS^®^	N/A
Cho et al., 2020	YES	N/A	N/A
Pichotano et al., 2019	NO	PRF + BIO-OSS^®^	BIO-OSS^®^
Pichotano et al., 2018	NO	PRF + BIO-OSS^®^ + COLLAGEN MEMBRANE	BIO-OSS^®^ + COLLAGEN MEMBRANE
Olgun et al., 2018	YES	N/A	N/A
Nizam et al., 2018	NO	PRF + BIO-OSS^®^	BIO-OSS^®^
Aoki et al., 2018	YES	N/A	N/A
Cömert Kılıç et al., 2017	NO	PRF + β-TCP/PRP + β-TCP	β-TCP ALONE
Aoki et al., 2016	YES	N/A	N/A
Gassling et al., 2013	NO	PRF + CORTILOCANCELOUS BONE + BIO-OSS^®^	CORTILOCANCELOUS BONE + BIO-OSS^®^ + COLLAGEN MEMBRANE
Zhang et al., 2012	NO	PRF + BIO-OSS^®^	BIO-OSS^®^
Tatullo et al., 2012	NO	PRF + BIO-OSS^®^	BIO-OSS^®^
Anitua et al., 2012	NO	PRF + BIO-OSS^®^	BIO-OSS^®^
Toffler et al., 2010	YES	N/A	N/A
Choukroun et al., 2006	NO	PRF + FREEZE-DRIED BONE ALLOGRAFT	FREEZE-DRIED BONE ALLOGRAFT

N/A = not available.

**Table 8 jcm-11-01888-t008:** Description of the amount of newly formed bone.

Author	New Bone Formation
Kempraj et al., 2020	PRF alone mean: 6.545 mm PRF + Bio-Oss^®^ 12.636 mm
Cho et al., 2020	2.5 mm ± 1.2 mm (PRF) 1.7 ± 1.0 mm (control)
Pichotano et al., 2019	PRF + DBBM 2.35 mm^2^ DBBM 1.58 mm^2^
Pichotano et al., 2018	Not reported
Olgun et al., 2018	Test 16.58 mm (1.05) Control 17.28 mm (2.53)
Nizam et al., 2018	Not reported
Aoki et al., 2018	Not reported
Kılıç et al., 2017	Mean of 33% in all groups
Aoki et al., 2016	Not reported
Gassling et al., 2013	PRF means 17% Collagen 17.2%
Zhang et al., 2012	Test group 18.35% ± 6.62% Control group 12.95% ± 5.33%
Tatullo et al., 2012	Not reported
Anitua et al., 2012	20–30% Test group 8% Control
Toffler et al., 2010	Mean increase 3.5 mm (3.4–5 mm)
Choukroun et al., 2006	Test group: 65% vital new bone 35% inert bone (4 months) Control group: 69% vital new bone 31% inert bone (8 months)

## Data Availability

All data are available within the study.

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
