# Peer review of "Sinus Lift Associated with Leucocyte-Platelet-Rich Fibrin (Second Generation) for Bone Gain: A Systematic Review"

_jcm, 2022, doi:10.3390/jcm11071888_

Round 1
Reviewer 1 Report
Manuscript ID: jcm-1616641
Title: Sinus lift associated with Leucocyte-Platelet Rich Fibrin (second generation) for bone gain: a systematic review
1.What is the main question addressed by the research?
To evaluate clinical studies on sinus floor elevation comparing the efficacy of platelet-rich fibrin (PRF) in bone regeneration whether alone or as a coadjutant to other bone graft materials.
2.Is it relevant and interesting?
The article is relevant and interesting.
3.How original is the topic?
The topic is current.
4.What does it add to the subject area compared with other published material?
The authors have collected and analyzed a great deal of recent data.
5.Is the paper well written?
Yes, the article is well written.
6.Is the text clear and easy to read?
Yes, but moderate English editing is required.
7.Are the conclusions consistent with the evidence and arguments presented?
Yes, the conclusions consistent with the evidence and arguments presented but further studies are needed to confirm these hypotheses.
8.Do they address the main question posed?
Yes, the Authors addressed the main question posed.
Other comments:
- English language: moderate English editing is required.
- Summary of abbreviations required.
- Introduction: This section needs some improvements. I would suggest inserting a sentence on academic debate on platelet concentrate effect: <<Efficacy of platelet concentrates in promoting wound healing and tissue regeneration is at the center of a recent academic debate [https://doi.org/10.1016/j.joms.2018.01.012]>>.
The Authors may improve this section also on the theme of crestal sinus lift with alternative instruments. Allow me to suggest a relevant references to include: “https://doi.org/10.3390/jpm12010108”.
- Materials and methods: This section has been properly prepared.
- Results: This section has been properly prepared.
- Discussion: A comparison with platelet-rich plasma is missing. Please improve.
- Conclusion: This section has been properly prepared but further studies are needed to confirm Authors’ hypotheses.
- Figures and Tables: Please improve figures and tables quality if possible.
After making the indicated changes, I am available for a second round of review.
Thanks for the opportunity to review this manuscript.
Author Response
- The article is relevant and interesting; The topic is current; The authors have collected and analyzed a great deal of recent data; The article is well written; The Authors addressed the main question posed; The conclusions consistent with the evidence and arguments presented but further studies are needed to confirm these hypotheses.
R: Thank you.
- Moderate English editing is required.
- Thank you. All the text was revised.
Other comments:
- Summary of abbreviations required. (R: It was included before References).
- Introduction: This section needs some improvements. I would suggest inserting a sentence on academic debate on platelet concentrate effect: <<Efficacy of platelet concentrates in promoting wound healing and tissue regeneration is at the center of a recent academic debate [https://doi.org/10.1016/j.joms.2018.01.012]>>.
The Authors may improve this section also on the theme of crestal sinus lift with alternative instruments. Allow me to suggest a relevant references to include: “https://doi.org/10.3390/jpm12010108”.
R: The introduction received new paragraphs according to recommendation.
- Materials and methods: This section has been properly prepared. (R: THANK YOU).
- Results: This section has been properly prepared. (R: THANK YOU).
- Discussion: A comparison with platelet-rich plasma is missing. Please improve. (R: A comparison between them was done and one article was cited).
- Conclusion: This section has been properly prepared but further studies are needed to confirm Authors’ hypotheses. (R: Thank you. We rewrote the conclusion).
- Figures and Tables: Please improve figures and tables quality if possible. (R: We tried to adjust some things in the tables to clarify them).
Reviewer 2 Report
Dear authors,
The study entitled" Sinus lift associated with Leucocyte-Platelet Rich Fibrin (second generation) for bone gain: a systematic review" explores a current approach in the clinical routine for dental surgeons. The study was very well conducted, following all the correct guidelines for systemic reviews. The studies selected present several differences between results and techniques applied. Then, the comparisons between the studies selected were compromised. The authors explained this very well using a chapter "limitations". However, the topic is quite interesting for the current dentistry, the results of this systematic review can not be extrapolated and are important to clarify this here. Almost no studies applied similar techniques, materials or investigations. Thus, the results are not totally comparable.
Below I suggest alterations to clarify the study and receive merit for publication.
1- Intro: The authors can discuss in the introduction section the historical approach of guided bone regeneration (GBR). This approach is usually applied for the same procedure and maybe more commonly applied in the clinical routine of dentists. I think the comparison is important for this study.
2- Results: The tables with "new bone formation" and "ISQ values" showed results using different techniques, materials and cases. I do not agree with these tables. The authors should divide the tables showing the studies that only used PRF alone and studies that used PRF with combinations. These results can not appear in the same table, they are not comparable for a final outcome such as new bone and ISQ.
3- Conclusions: The conclusions section, should describe again the study limitations. I agree with the conclusions described, however, the systematic review proposed here doesn't demonstrate a consistent final outcome. For while, the conclusion section is a combination of different conclusions from the studies. The authors should describe that is necessary more studies to create a significant conclusion.
Author Response
Reviewer 2
Then, the comparisons between the studies selected were compromised. The authors explained this very well using a chapter "limitations". However, the topic is quite interesting for the current dentistry, the results of this systematic review can not be extrapolated and are important to clarify this here. Almost no studies applied similar techniques, materials or investigations. Thus, the results are not totally comparable.
R: We agreed and rewrote the conclusion.
1- Intro: The authors can discuss in the introduction section the historical approach of guided bone regeneration (GBR). This approach is usually applied for the same procedure and maybe more commonly applied in the clinical routine of dentists. I think the comparison is important for this study.
R: The introduction received new paragraphs according to recommendation.
2- Results: The tables with "new bone formation" and "ISQ values" showed results using different techniques, materials and cases. I do not agree with these tables. The authors should divide the tables showing the studies that only used PRF alone and studies that used PRF with combinations. These results can not appear in the same table, they are not comparable for a final outcome such as new bone and ISQ.
R: The tables were adjusted as requested.
3- Conclusions: The conclusions section, should describe again the study limitations. I agree with the conclusions described, however, the systematic review proposed here doesn't demonstrate a consistent final outcome. For while, the conclusion section is a combination of different conclusions from the studies. The authors should describe that is necessary more studies to create a significant conclusion.
R: Thank you. We rewrote the conclusion to correctly achieve the comments.
Round 2
Reviewer 1 Report
Authors' changes improved manuscript quality.
Anyway, introduction section still need some improvements.I would suggest inserting a sentence on academic debate on platelet concentrate effect: <<Efficacy of platelet concentrates in promoting wound healing and tissue regeneration is at the center of a recent academic debate [https://doi.org/10.1016/j.joms.2018.01.012]>>.
The Authors may improve this section also on the theme of crestal sinus lift with magneto-dynamical technology using the following reference: “https://doi.org/10.3390/jpm12010108”.
Author Response
"Anyway, introduction section still need some improvements.I would suggest inserting a sentence on academic debate on platelet concentrate effect: <<Efficacy of platelet concentrates in promoting wound healing and tissue regeneration is at the center of a recent academic debate [https://doi.org/10.1016/j.joms.2018.01.012]>>.
The Authors may improve this section also on the theme of crestal sinus lift with magneto-dynamical technology using the following reference: “https://doi.org/10.3390/jpm12010108”."
R.: Thank you. The introduction was improved as requested.
Reviewer 2 Report
The authors corrected with perfection the suggested comments.
Now, the conclusion section provides information in accordance with the study performed. Moreover, this review can improve the knowledge in this topic and stimulate new studies for more consistent results.
Author Response
"The authors corrected with perfection the suggested comments.
Now, the conclusion section provides information in accordance with the study performed. Moreover, this review can improve the knowledge in this topic and stimulate new studies for more consistent results."
R.: Thank you so much for the words.